# Study on the Pre-Oxidation and Resulting Oxidation Mechanism and Kinetics of Mo-9Si-8B Alloy

**DOI:** 10.3390/ma14185309

**Published:** 2021-09-15

**Authors:** Cheng Wang, Qiuliang Li, Zhenping Guo, Xiangrong Li, Xiangyu Ding, Xin Li, Zhuoyue Li, Bin Li

**Affiliations:** 1Fundamentals Department, Air Force Engineering University, Xi’an 710038, China; qiuliang_95@163.com (Q.L.); pluto_70@126.com (Z.G.); lixiangrong0925@126.com (X.L.); dingxiangyu@nchu.edu.cn (X.D.); lx202006t10@163.com (X.L.); lz_980512@163.com (Z.L.); 15770518701@163.com (B.L.); 2School of Aircraft Engineering, Nanchang Hangkong University, Nanchang 330063, China

**Keywords:** pre-oxidation, molybdenum silicon boron alloy, hinge locking

## Abstract

Molybdenum silicon boron alloy is regarded as the next generation of superalloy that is expected to replace nickel-based superalloys. However, the high-temperature oxidation resistance of Mo-Si-B alloy has always been an issue worth studying. In this study, Mo-9Si-8B alloy was prepared via a plasma oscillatory pressure sintering process and pre-oxidized at 1300 °C while maintaining a certain balance of mechanical and oxidation properties. The influence of the oxide protective layer on its performance at high temperature of 1150 °C was explored, the micro-mechanism of its performance and its failure mode of the hinge-locking mechanism was illustrated, and finally, its oxidation kinetics was inferred. In conclusion, pre-oxidized Mo-9Si-8B (at.%) alloy did play a role in delaying the oxidation process during the initial period of cyclic oxidation. However, with the increase of cyclic oxidation time, the improvement of high-temperature oxidation resistance was limited.

## 1. Introduction

For existing nickel-based superalloys, they could be used at around 900 °C without the coating and cooling system. Even with the addition of the coating and cooling system, the nickel-based alloy could only work at about 1150 °C [1], which had reached its maximum limit and resulted in a huge reduction in the efficiency of the turbine engine. The use of non-cooled engine components will greatly reduce the amount of cooling air and increase thrust. The calculations in Figure 1a show that for every 1% reduction in nozzle guide vane (NGV) cooling gas, the engine thrust increases by 0.84%, the engine fuel flow increases by 1.43%, and the specific fuel consumption (SFC) increases by 0.54%. Mo-Si, Nb-Si, and Si-C ceramic matrix composites were the main candidates for high-temperature parts of aero-engines in the future [2]. The specific strength of the Si-C ceramic matrix composites was low, and its intrinsic brittleness was too large. It was not suitable for load-bearing parts of complex shapes in high-temperature circumstances, and it was difficult to connect ceramic matrix composites with metal materials [3]. The specific strength and oxidation resistance of Nb-Si alloy above 1300 °C were obviously weaker than that of Mo-Si alloy [1,2,3,4]. It had been difficult to meet the high-temperature resistance requirements of high-temperature parts [5]. Mo-Si-B alloy as one of the Mo-Si alloys was the attractive material considered as the key direction of the future development of superalloys, which had been explored for decades. Due to its remarkable properties, it had been expected to be an alternative for next-generation turbine engine blades. The Mo-Si-B alloy exhibited outstanding mechanical properties, high-temperature oxidation resistance, good creep resistance, and high melting point. However, it is hard for the Mo-Si-B alloy to keep both the mechanical properties and high-temperature oxidation resistance excellent. While maintaining certain mechanical properties, it is still a problem to continue to maintain and improve the oxidation resistance. There were several methods of approaching this problem: (1) The first method was through microalloying, doping with trace elements to form stable oxides, reducing the diffusion rate of oxygen to the internal matrix, and improving the adhesion of the oxide layer. Pogozhev found that adding a small amount of Hf can enhance the wettability of the oxide layer [6], and it played a role in changing the microstructure of the oxide film, improving the toughness of the oxide film, and preventing the oxide film from cracking [7]. (2) The second way was to design a suitable Si/B ratio. Supatarawanich et al. [8] had found that Mo-Si-B alloys had better oxidation resistance at 1300 °C with the Si/B ratio close to 1. (3) The third way was to improve the oxidation resistance of the alloy by refining the grain, but as the grain boundaries increase, Si was more likely to segregate to the grain boundary, causing the strength of the grain boundary to decrease [9]. (4) At last, the pre-oxidation method was also reckoned as one of the potential ways to solve this problem.

## 2. Materials and Methods

The samples were prepared from elemental powder mixtures of Mo, Si, and B of 99.95 wt.%, 99.99 wt.%, 99.95 wt.% purity with an average particle size of 2.0–3.5 µm, 3.0–5.0 µm, and 0.5–1.0 µm, respectively. The alloy composition chosen for this study was located within the three phases field between α-Mo + Mo_5_SiB_2_ + MoSi_3_ (Figure 1c). All samples were processed by a powder metallurgical (PM) route. The mixed powders were put into the planetary mill (QM-3SP04, Nanjing Laibu Technology Industrial Co., LTD, Nanjing, China) with a speed of 300 rpm and a powder-to-ball weight ratio of 1:10. The ball mill rotated forward and reversed for 12 h respectively to get a uniform mixture. At the same time, it also included a plasma oscillation sintering process, which adopted 6 h heat preservation time and a regular pressure oscillation curve, keeping the frequency at 9 Hz (periodic change). In addition, the oscillatory sintering furnace independently designed by Morningstar Weike Co., Ltd. (Ningbo, China) can greatly increase the density of the alloy.

The pre-oxidation behavior of the Mo-9Si-8B alloy was investigated. The pre-oxidation specimens were prepared by cutting cylindrical samples with dimensions of Φ20 × 5 mm Then, the cylindrical samples were grounded with SiC paper to a grit of 2000 and polished with 1 and 0.25 μm diamond paste. Next, the samples were pre-oxidized at 1300 °C under 10^−6^ Pa oxygen partial pressure for 2 h (Figure 1b). At last, the cyclic oxidation experiment was implemented. The cyclic oxidation experiment was to keep the temperature at 1150 °C for 50 h to verify the effectiveness of the pre-oxidation. Samples used for the oxidation test were taken out to weigh at 0.5 h, 1 h, 5 h, 10 h, 20 h, 30 h, and 50 h, respectively, and cooled to room temperature in air at the time of each weight.

The microstructure was etched using the Murakami’s etch (an aqueous solution of 10 vol.% potassium ferricyanide and 10 vol.% sodium hydroxide) before using the scanning electron microscope (SEM, TESCAN, Brno, Czech Republic). Meanwhile, element diffusion and distribution were explored by energy dispersion spectroscopy (EDS, Oxford Instruments Co., Oxford, UK). After the oxidation, the constituent phases and structure of the oxidation scale were identified by X-ray diffraction using the Cu-K_α_ radiation (XRD, XRD7000S, Shimadzu, Tokyo, Japan).

## 3. Results and Discussion

### 3.1. Microstructures of the Plasma Oscillation Sintered Composites

Figure 2a showed the surface microstructure of the sintered alloy, and the high magnification was revealed in Figure 2b. The continuous Mo_ss_ (light phase, Mo solid solution), Mo_5_SiB_2_ (dark phase), and Mo_3_Si (gray phase) were confirmed by the XRD result, and the distribution of these phases was inferred by EDS scanning. The intermetallic phase can be seen clearly and dispersed in the Mo_ss_ base phase. The area ratio was used to replace the volume fraction, and the average volume fraction of Mo_ss_ phase was 54.9 vol.% calculated 10 times by Image-Pro Plus software (6.0). To some extent, the volume fraction of the Mo_ss_ phase was close to the expected target of 50 vol.%, which committed to maintaining relatively balanced mechanical properties and antioxidant properties [12,13,14]. The density increase of oscillatory sintering can be measured by the Archimedes suspension wire method. Compared with traditional hot-pressing sintering, this density was increased by 9.25 g/cm^3^, and its relative density reached 97.78%.

### 3.2. Cross-Section of Pre-Oxidation Alloy

As shown in Figure 3a,c, after 30 min of pre-oxidation, a discontinuous oxide layer was formed on the outermost surface of the alloy (Figure 3a,c). Under the outermost oxide layer, an intermediate layer of 30.6 μm existed on the substrate (Figure 3c). After investigation and analysis by EDS, the black granular part was SiO_2_ oxide, and the rest was almost the Mo_ss_ phase. The region ‘A’ is the region that firstly underwent an oxidation reaction to form a large amount of silica, which can be confirmed by the EDS scanning results marked ‘A’ in Figure 3b. The formation of silicon dioxide consumed the intermetallic phase, causing the unoxidized part of the outermost layer to belong to the Mo_ss_ phase, which was consistent with Mendiratta’s results [15]. Another point worth noting was that in the first oxidized area ‘A’, the reduction of the distribution of boron can be clearly found, which indicated that the boron oxide volatilized rapidly at this temperature, leading to the loss of B element. Figure 3d,e showed the morphology of the surface oxide layer after 1 h. It could be seen that a ≈1–2 μm thin containing-SiO_2_ layer had been formed. The most important point was that in the contact part between the outermost oxide and the intermediate layer, there were black silicon dioxide particles grown from the substrate to link the oxide layer with the intermediate layer, which acted as a mechanical locking [16]. It closely connected the SiO_2_ oxide layer with the substrate, which reinforced the protection. After 2 h of pre-oxidation as observed in Figure 3f, the depth of mechanical locking was further deepened, and a dense silica protective layer was formed on the surface of the substrate, which adhered well to the substrate.

### 3.3. EDS Analysis and Surface Morphology of the Pre-Oxidized Alloy

Figure 4 showed the surface morphology of the sample after 2 h of pre-oxidation, as observed in the secondary electron and backscatter modes, respectively. It can be seen from Figure 4a that a dense layer with no bubbles completely covered the surface of the substrate. We have selected five spots to do EDS, and all these spots have similar results. A typical result of these spots is shown in Figure 4b. From the analysis of the EDS results, it can be concluded that there may exist a certain amount of boron oxide and silicon dioxide under this condition. In addition, under this condition, almost no Mo content appeared on the surface, which implied that there was almost no molybdenum oxidation on the surface. After 2 h of pre-oxidation, the surface of the sample formed a dense borosilicate layer.

### 3.4. Oxidation Behavior of Mo-Si-B Alloy at 1150 °C

In this study, the protective effect of the pre-oxidation layer at 1150 °C was discussed here. The engine starting process was similar to the cyclic oxidation, so the cyclic oxidation at 1150 °C was selected. The surface and cross-sectional morphology of cyclic oxidation at 1150 °C after 50 h of oxidation is shown in Figure 5. Figure 5a showed the appearance of holes, cracks, and protrusions on the surface. At the same time, Figure 5b contained a large number of cracks and there even existed oxide-scale spalling. The existence of the pores may be due to the volatilization of molybdenum oxide inside, which caused the bubbles to rush out of the external oxide layer and remain holes. In an air atmosphere, the higher oxygen concentration can no longer maintain the selective oxidation conditions of silicide and borides. Due to a large amount of Mo element, oxygen reached the oxide/matrix interface and reacted with Mo atoms to form MoO_3_ and MoO_2_. The formation of MoO_3_ was quickly lost in the form of gas, resulting in the rupture of the borosilicate protective layer on the surface, further accelerating the process of oxygen invasion. The cyclic oxidation process had undergone multiple heating/cooling processes, which can be explained according to Plling’s classical “*PBR*” theory(Pilling\|Bedworth Ratio) [17]. In the process of metal forming oxide, the volume changes to produce growth strain, which is defined as “*PBR*”:(1)$$PRB=\frac{V_{OX}}{V_{M}}$$
where *V_ox_* is the volume of oxide formed, and *V_M_* is the volume of metal consumed.

The oxide film expanded horizontally and vertically, and it was divided into horizontal growth and vertical growth. Since the thermal expansion coefficient of the oxide layer and the substrate were different, the degree of volume expansion was inconsistent, which will cause internal stress. When *PBR* > 1, compressive stress was generated in the oxide film; when *PBR* < 1, tensile stress was generated in the oxide film [18]. In this study, the *PBR* of Mo_3_Si to SiO_2_ of the pre-oxidized alloy is about 2.1, and the *PBR* of Mo_5_SiB_2_ to SiO_2_ and B_2_O_3_ is about 1.01 based on the oxide layer formed by the pre-oxidation and the volume of metal consumed, which can refer to Li’s calculation method [18]. As a result, the pre-oxidized film generated compressive stress, and the subsequent cyclic oxidation further led to inconsistent expansion and contraction rates. The oxide layer shrunk and expanded less than the substrate. Therefore, the cooling process of the oxide layer was bound by the substrate. When the internal stress was greater than the critical stress that the oxide layer can withstand, the stress will cause cracks and spalling behavior. Furthermore, as shown in Figure 3, the mechanical locking mechanism played a very good ‘hinge-locking’ function at first, which held the borosilicate layer tightly on the surface of the substrate, but as the internal oxide changed from molybdenum-based to molybdenum oxide in Figure 4d and Figure 5e, we suspected it was its coefficient of thermal expansion changing from 6 × 10^−6^ K^−1^ to 5 × 10^−6^ K^−1^ that resulted in the reduction of the transverse pressure on the SiO_2_ embedded in the substrate. As the MoO_3_ rushed out in the form of bubbles, a loose MoO_3_ area was formed, which further released the lateral pressure on SiO_2_ and gradually led to the failure of the mechanical locking mechanism.

As shown in Figure 6a, the oxidation mass loss curve at 1150 °C was plotted. The pre-oxidized sample had a mass loss of −107.5 mg/cm^2^ after 50 h of exposure to the air, which was −198.3 mg/cm^2^ compared to the sample without pre-oxidation treatment. Although the weightlessness has been reduced by nearly 91 mg/cm^2^, the weightlessness of 50 h had not been substantially improved. However, it can be seen from the partially enlarged view of Figure 6b that the pre-oxidation layer played a very good protective role in the initial period and delayed the progress of the oxidation behavior.

According to the mass loss trend, the curve was divided into three parts (Figure 6). At the initial stage, the mass loss of the pre-oxidation samples was significantly lower than that of the untreated samples, which indicated that the dense oxide layer effectively reduced the rate of oxygen diffusion. In the subsequent stage, the mass loss of the pre-oxidized samples increased significantly, indicating that under the condition of high oxygen partial pressure, the thinner protective layer formed by the pre-oxidation could not completely prevent the intrusion of oxygen. The mass loss rate of MoO_3_ was faster than the formation of the borosilicate layer, showing linear weight loss. When the oxidation time exceeded 30 h, the pre-oxidation samples had a severe mass loss. However, the alloy experienced a mass loss with −14 mg/cm^2^ during the first 10 h, which was relatively better than Rioult’s data of −17 mg/cm^2^ mass loss in 10 h at 1100 °C [19,20]. It was worth pointing out that in the first 5 h earlier, the alloy experienced a mass loss of −1.8 mg/cm^2^, which was better than Helmick’s data of −40 mg/cm^2^ [21]. Hence, it exhibited an improved oxidation resistance in the initial stage at 1150 °C.

### 3.5. Inferred Analysis and Mechanisms

The formation of the oxide layer includes two parts: the diffusion process and the chemical reaction. The diffusion process requires a chemical potential gradient as a driving force. In the chemical reaction process, the chemical reaction proceeds in the direction of the decrease of Gibbs free enthalpy. The pre-oxidation process is carried out at 1300 °C, so the Gibbs free energy of each phase reacting with 1 mol of oxygen is calculated. The calculation of Gibbs free energy usually includes two calculation methods: one uses the standard reaction heat effect calculation, and the other uses the standard reaction difference entropy calculation. In this study, a simplified algorithm is used: the material Gibbs free energy function method [22,23]. The algorithm is based on the classic algorithm without any assumptions, and the result is exactly the same as the classic algorithm.
(2)$$\Delta G_{T}^{0}=\Delta H_{T_{0}}^{\theta }-T\Delta \varphi_{T}^{'}$$
where $\Delta H_{T_{0}}^{\theta }$ is the standard enthalpy of reaction at *T*_o_, $\Delta \varphi_{T}^{'}$ is the Gibbs free energy function of the reaction, $\Delta G_{T}^{0}$ represents the Gibbs free energy of the reaction.

First, the Gibbs free energy function of the reaction at 1300 °C (1573 K) (Δ*φ*’_1573_) is calculated by the following linear Equation (4). All the results are shown in Table 1.
(3)$$\Delta \varphi_{1573}^{'}=\Delta \varphi_{1500}^{'}+\frac{\Delta \varphi_{1600}^{'}-\Delta \varphi_{1500}^{'}}{1600-1500}\left(1573-1500\right)$$

Second, under 1300 °C and different oxygen partial pressure, the following three kinds of reaction equations may exist:

In the case of high oxygen partial pressure:(4)23Mo+O2→23MoO3
(5)211Mo3Si+O2→611MoO3+211SiO2
(6)110Mo5SiB2+O2→12MoO3+110SiO2+110B2O3.

In the case of further reduction of oxygen partial pressure:(7)$$Mo+O_{2}\to MoO_{2}$$
(8)14Mo3Si+O2→34MoO2+14SiO2
(9)110Mo5SiB2+O2→12MoO3+110SiO2+110B2O3.

In the case of low oxygen partial pressure:(10)$$Mo_{3}Si+O_{2}\to 3Mo+SiO_{2}$$
(11)25Mo5SiB2+O2→2Mo+25SiO2+25B2O3.

Third, the diffusion path and the oxidation products of each equation are determined by the oxygen partial pressure at the oxide/matrix interface. The oxygen partial pressure of each equation is calculated according to the following Equation (12):(12)$$logP_{O_{2}}=\Delta G_{T}^{\theta }/2.303RT$$
where *R* is the gas constant, and $P_{O_{2}}$ represents the equilibrium oxygen partial pressure.

According to the above data and equations, the Gibbs free energy and oxygen partial pressure of each phase of Mo-Si-B alloy reacting with 1 mol oxygen is calculated as follows.

It can be seen from Table 2 that the Gibbs free energy of Equation (6) and Equation (7) is less than that of Equation (5), and similarly, the Gibbs free energy of Equation (9) and Equation (10) is also less than that of Equation (8). It is all pointed that the oxidation behavior of the intermetallic phase is before the Mo_ss_ phase at three levels of oxygen partial pressure. In particular, the oxygen partial pressure required for the reaction of Equation (5) is at the level of 1.10 × 10^−6^, which is slightly higher than the level of designed ambient oxygen partial pressure. It indicates that the activity of generating MoO_3_ is inhibited from the beginning and proceeds at a very slow rate; then, it evaporates instantly at a high temperature of 1300 °C. The outermost intermetallic phase is oxidized to produce SiO_2_ melt and B_2_O_3_. Although the viscosity of SiO_2_ decreases at high temperatures, it is still as high as 10^9^ dpa·s. As shown in Figure 7a, in the early stage of pre-oxidation, the B source provided from the substrate is divided into two parts: one part evaporates directly from the substrate, and the other part enters the formed SiO_2_ melt. B_2_O_3_(g) that existed in the SiO_2_ melt gradually increases from the bottom to the top due to changes in pressure. The volume change promotes the liquid up and down convection; the melt SiO_2_ collapses and flows after blasting. Therefore, the presence of B_2_O_3_ increases the flow force of high-viscosity SiO_2_ melt and accelerates the covering process. Then, most of B_2_O_3_ escapes from the borosilicate layer at a very fast evaporation rate, leaving a high-viscosity SiO_2_ layer, reducing the oxygen diffusion rate by six orders of magnitude [24]. With the increase of the depth of the matrix, the partial pressure of oxygen is further reduced, and the selective oxidation of Si occurs. The intermetallic phase is consumed to provide the Si/B source [25]. After two hours, an intermediate layer of Mo_ss_ distributed with SiO_2_ granular is formed.

As shown in Figure 7b, the B-poor borosilicate layer formed by pre-oxidation has a mechanical locking effect and plays a role of articulation. According to the calculated PRB values of 2.1 > 1 and 1.01 > 1, the SiO_2_ formed by the first selective oxidation will expand along with the lateral growth and the longitudinal growth. Since the longitudinal volume expansion hardly generates stress, the inwardly extending SiO_2_ particles undergo lateral expansion to generate compressive stress. The pre-oxidized intermediate layer is the Mo-based layer, and the thermal expansion coefficient is 10 times larger than the SiO_2_ melt. The lateral volume expansion causes stronger compressive stress, forming a locking mechanism, increasing the adhesion to the substrate.

The pre-oxidized samples were cyclically oxidized at 1150 °C for 50 h. The parabolic mass loss proved to be protective for the initial period, but then, it changed to linear weight loss, and the protective layer became ineffective. One is because MoO_2_ and MoO_3_ are generated with the increase of oxygen partial pressure and the inward diffusion of oxygen. MoO_3_ evaporated, forming small bubbles, and it escapes the borosilicate layer, causing holes and flakes in the protective layer [26,27,28]. The pores make MoO_3_ a loose structure and become a channel for oxygen diffusion and further intrusion. Then, the internal matrix appeared as a large mass loss. On the other hand, it is caused by the inconsistent thermal expansion coefficient of the protective layer. The thermal expansion coefficient of the produced MoO_3_ and MoO_2_ (5 × 10^−6^ K^−1^) intermediate layer is lower than Mo_ss_ (6 × 10^−6^ K^−1^), which weakens the volumetric compressive stress of SiO_2_ particles with a ‘mechanical locking’ effect. The loose structure of MoO_3_ and MoO_2_ further released the compressive stress, as shown in Figure 7c, resulting in the loss of the SiO_2_ mechanical locking function that functions as a ‘hinge locking’. At last, the adhesion of the protective layer decreases, spalling occurs, and the protective layer fails, which confirms the results in Figure 4. At last, the mechanism diagram of the pre-oxidation of Mo-Si-B alloy and the cyclic oxidation at 1150 °C is illustrated in Figure 7, which is contributed to understand the mechanism of pre-oxidation behavior of Mo-Si-B alloy.

## 4. Conclusions

In this paper, the pre-oxidation of the Mo-9Si-8B alloy was carried out under the conditions of 1300 °C and 10^−6^ Pa oxygen partial pressure. The microstructure evolution process and the pre-oxidation mechanism were illustrated and explained, and then the cyclic oxidation was carried out at 1150 °C for 50 h. The oxidation kinetics was discussed here, and the following conclusions were drawn:

Pre-oxidation formed a borosilicate layer with a thickness of 3–5 μm. The selective oxidation of Si resulted in the formation of SiO_2_, and its lateral growth produced compressive stress, which led to a ‘hinge-locking’ mechanism and increased the adhesion to the substrate.In the pre-oxidation stage, part of B_2_O_3_ escaped directly from the substrate, and the other part entered the SiO_2_ layer. Its volume change promoted liquid flow and the movement of the bubbles from the bottom to top promoted the convection of SiO_2_ melt. Eventually, blasting caused SiO_2_ liquid to collapse and flow, which improved the fluidity of the protective layer.The reason for the failure of the ‘hinge-locking’ mechanism in the cyclic oxidation stage was that the MoO_3_ and MoO_2_ intermediate layer replaced the position of the Mo_ss_ phase, and the thermal expansion coefficient of MoO_3_ and MoO_2_ intermediate layers decreased, which led to the decrease of the pressure on the SiO_2_ particles with the locking effect. In addition, the loose structure caused by MoO_3_ bubbles further released the pressure stress, resulting in the failure of the ‘link-locking’ mechanism.Pre-oxidation played a role in delaying the oxidation process in the initial stage of the cyclic oxidation, but as the cycle oxidation time increased, its protective effect was lost and entered the rapid weight loss stage, but the 10 h oxidation result still had certain advantages.

## Figures and Tables

**Figure 1 materials-14-05309-f001:**
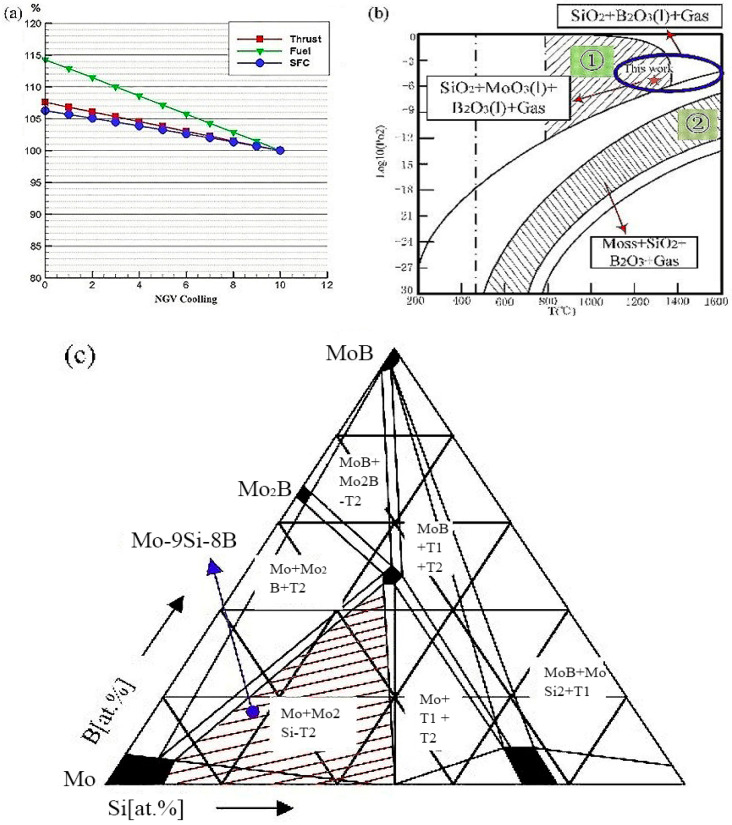
(**a**) Effect of high-conductivity cooling air volume on engine performance in our calculation: Thrust is getting higher with the decrease of the cooling air volume. (**b**) Calculated oxygen partial pressure diagram for the Mo-9Si-8B-O system Adapted from [10]. (**c**) Isothermal section of Mo-Si-B at 1600 °C, the composition chosen for this study was marked Adapted from [11].

**Figure 2 materials-14-05309-f002:**
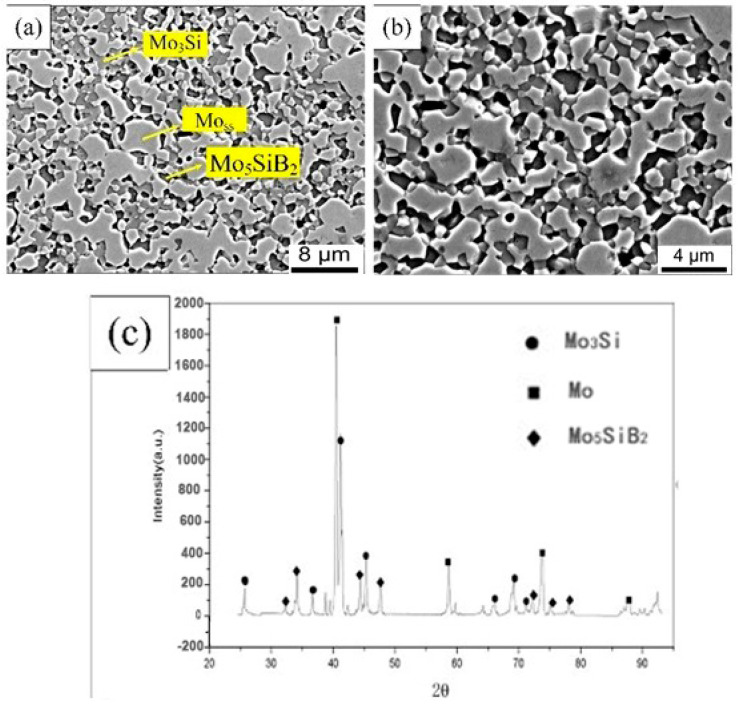
Microstructure of alloy prepared by plasma concussion sintering: (**a**) 3000×; (**b**) 5000×; (**c**) X-ray diffraction pattern.

**Figure 3 materials-14-05309-f003:**
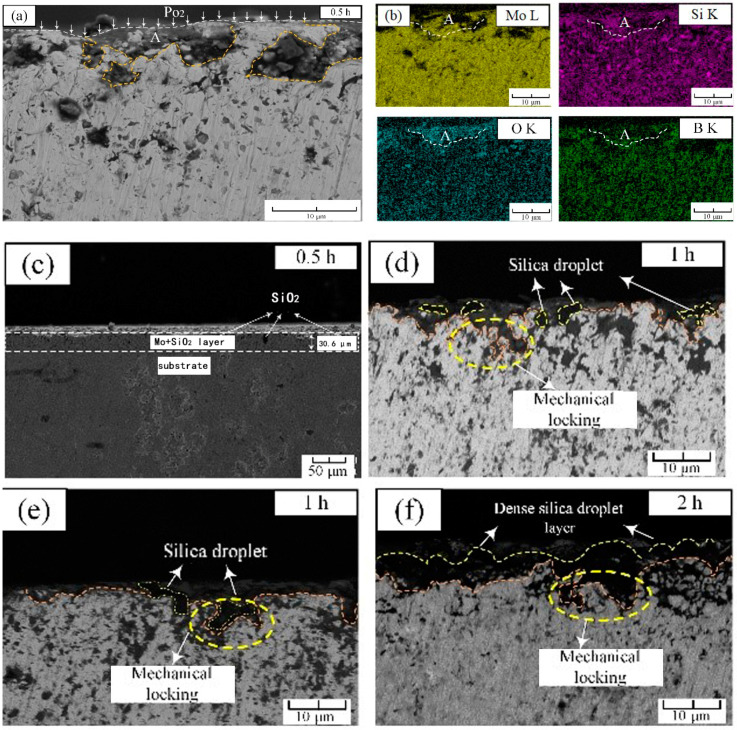
Cross-sectional morphology of pre-oxidized alloy at 1300 °C: (**a**) pre-oxidation for 0.5 h; (**b**) local EDS mappings in (**a**); (**c**) macro-image of (**a**) for 30 min; (**d**,**e**) pre-oxidation for 1 h; (**f**) pre-oxidation for 2 h.

**Figure 4 materials-14-05309-f004:**
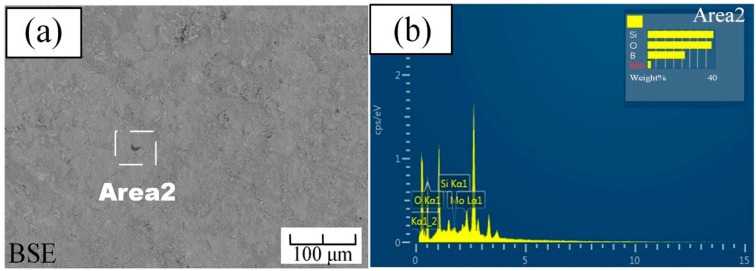
Pre-oxidized surface of the Mo-9Si-8B alloy for 2 h: (**a**) backscattered electron image; (**b**) EDS of Area 2 in (**a**).

**Figure 5 materials-14-05309-f005:**
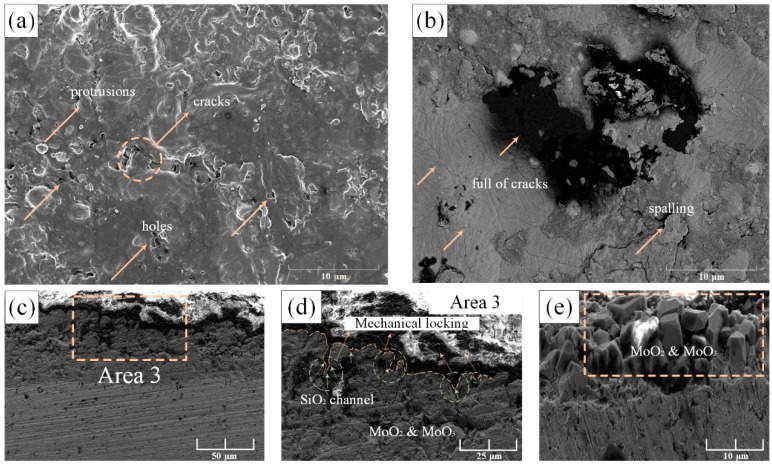
SEM microstructures of pre-oxidized Mo-9Si-8B composite after oxidation in air at 1150 °C for 50 h: (**a**,**b**) are surface SEM images of the alloy; (**c**) cross-sectional morphology of the edge; (**d**) local magnification of area 3 in (**a**); (**e**) molybdenum oxide products grown from the substrate.

**Figure 6 materials-14-05309-f006:**
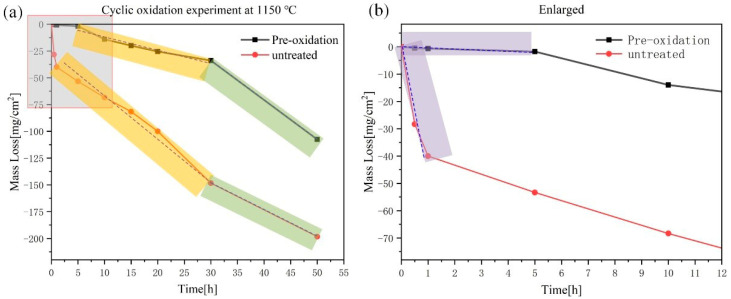
Oxidation kinetic curve at 1150 °C for 50 h in air: (**a**) oxidation mass loss curves of pre-oxidized and untreated samples; (**b**) local enlarged part in (**a**).

**Figure 7 materials-14-05309-f007:**
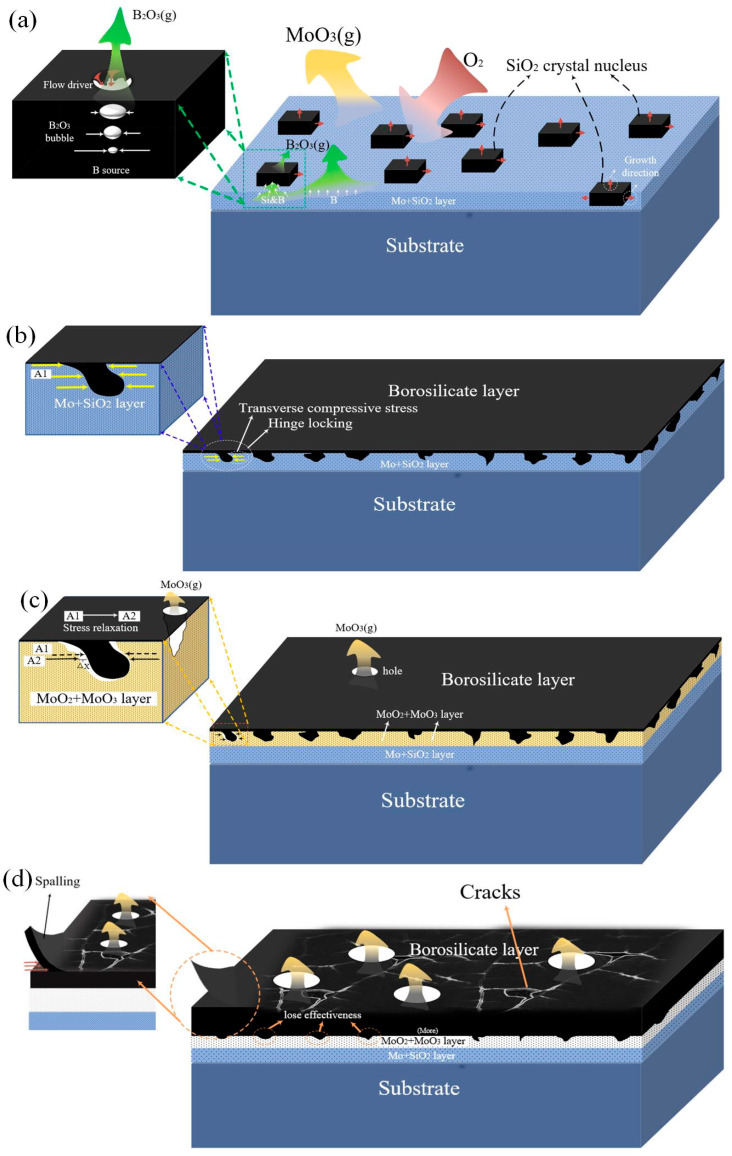
Pre-oxidation and cyclic oxidation mechanism diagram:(**a**,**b**) are pre-oxidation progress at 1300 °C; (**c**,**d**) are cyclic oxidation progress at 1150 °C.

**Table 1 materials-14-05309-t001:** Thermodynamic data of some materials.

Species	$\Delta H_{298}^{\theta }\;\left(J/mol\right)$	$\varphi_{T}^{'}\;\left(1500\;K\right)$	$\varphi_{T}^{'}\;\left(1600\;K\right)$	$\Delta S_{298}^{\theta }\;\;J/\left(mol\cdot K\right)$	$\varphi_{T}^{'}\;\left(1573\;K\right)$
Si	0	37.991	39.325	18.828	38.965
SiO_2_	−847,260	94.535	98.027	46.861	97.084
B_2_O_3_	−1,252,188	160.427	166.642	78.404	164.964
Mo	0	50.011	51.467	28.606	51.074
O_2_	0	230.835	232.579	205.016	232.108
Mo_3_Si	−116,399	187.129	192.698	106.148	191.194
Mo_5_SiB_2_	−309,616	369.280	380.389	207.342	377.389
MoO_2_	−587,852	106.092	110.217	49.999	109.103
MoO_3_	−360,661	338.169	342.156	279.910	341.079

**Table 2 materials-14-05309-t002:** Values of $\Delta G_{T}^{\theta }$ and $P_{O_{2}}$ of the Equation (4) to Equation (11).

**Equations**	Equation (5)	Equation (6)	Equation (7)	Equation (8)
$\Delta G_{T}^{\theta }\left(J/mol\right)$	−179,453.39	−230,225.86	−244,327.40	−314,025.73
$P_{O_{2}}$(Pa)	1.10 × 10^−6^	2.27 × 10^−8^	7.72 × 10^−9^	3.75 × 10^−11^
**Equations**	Equation (9)	Equation (10)	Equation (11)	Equation (12)
$\Delta G_{T}^{\theta }\left(J/mol\right)$	−350,203.87	−355,661.54	−458,738.29	−438,933.16
$P_{O_{2}}$(Pa)	2.36 × 10^−12^	1.55 × 10^−12^	5.77 × 10^−16^	2.67 × 10^−15^

## Data Availability

The data presented in this study cannot be shared at this time as the data also forms part of an ongoing study.

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
