# Peer review of "Study on the Pre-Oxidation and Resulting Oxidation Mechanism and Kinetics of Mo-9Si-8B Alloy"

_materials, 2021, doi:10.3390/ma14185309_

Round 1
Reviewer 1 Report
The authors have well explained and analysed the study on Pre-oxidation and Resulting Oxidation mechanism and kinetics of Molybdenum silicon boron alloy. The following are the points that must be considered before publication of manuscript.
- The figures are not very clear, please make the figures properly for the easiness of readers.
- In the line 106, Fig.3(g) is discussed but this figure is not mentioned in Fig.3
Author Response
Dear Editor Sylvia Wu and Reviewers,
On behalf of my co-authors, we thank you very much for giving us an opportunity to revise our manuscript. Thanks very much for taking your time to review this manuscript. I really appreciate all your comments and suggestions! Please find my itemized responses in below and my revisions in the re-submitted files.
Thanks again!
Sincerely,
Cheng Wang

Reviewer 2 Report
Paper: Study on the Pre-oxidation and Resulting Oxidation Mechanism and Kinetics of Mo-9Si-8B Alloy,
Comments:
Abstract
Please re-write these phrases, there is something not write between them: Molybdenum silicon boron alloy is regarded as the next generation of superalloy that is expected to replace nickel-based superalloys. However, the high temperature oxidation resistance of Mo-Si-B alloy has always been a worth studying issue.
The influence of the protective layer on its performance at a high temperature of 1150oC
Does- protective layer means a coating or oxide scale?
Introduction
The Mo-Si-B alloy exhibited outstanding mechanical properties, high-temperature oxidation resistance, good creep resistance, and high melting point – so why you are doing this research since all is excellent?
The use of non-cooled engine components will greatly reduce the amount of cooling air and increase thrust as shown in Fig.1(a) – why this figure is not shoed just underneath the sentence?
Mo-Si, Nb-Si, and Si-C ceramic matrix composites were the main candidates for high-temperature parts of aero-engines in the future – any reference, why do authors think like that?
The specific strength and oxidation resistance of Nb-Si alloy above 1300℃ were obviously weaker than that of Mo-Si alloy – why were weaker?
In the next sentence oxidation resistance may be improved – so oxidation resistance is outstanding or must be improved? Please re-write this part of the paper.
Page 2, line 49: do not start the new sentence from and: And it played a role in changing
Page 2, line 52, please re-write this sentence as according to research performed by Supatarawanich, please add a reference
Page 2, 52: what means: under the high-temperature environment – if you talk about high-temperature environment – please add some more details
Materials and Methods
What stands for the circular bulks – is the sample? If yes, please rename: cylindrical sample
Page 2, line 77: add more details: were mechanically ground and polished.
Page 2, line 78: oxygen partial pressure for 2 h – it was pure oxygen? If yes please add this information.
The cyclic oxidation experiment was to keep the temperature at 1150℃ for 50 h to verify the effectiveness of the pre-oxidation. The cyclic oxidation experiment was weighed at 0.5h, 1h, 5h, 10h, 20h, 30h, 50h, respectively – this is not cyclic oxidation. Cyclic oxidation means you have an equal cycle at high temperature. Please re-write this sentence.
Figure 2 A, B, C, Figures must be much bigger it is hard what authors found, X-ray diffraction pattern not image
EDS cannot confirm phases, EDS can only show chemical composition, XRD may do it
Moss – is that the molybdenum substrate
Page 3, line 100: and its relative density reached 97.78% - is this was the final density ?
Page 3, line 108, was SiO2 oxide - subscript
Page 3, line 109: firstly underwent an oxidation reaction to form massive silica – what is massive silica ?
Figure 3, it should only cross-section after 2h of pre-oxidation process, no need to add SEM images after: 0.5, 1h. The SEM image after 0.5h shows not much to be honest.
What is different in images 3A and 3C are both are after the 0.5 h pre-oxidation process?
Figure 4, no point to add EDS spectra shows nothing special to a whole paper, what EDS results show to a reader? Just nothing, just peaks.
No need to add SE and BSE images, just add BSE image SE image must be removed, if you are making EDS please make sure you are doing as least 4 – 5 spots, here a single spot was done not much
Page 4, line 141: No capital letter after coma
There is loads of issues in this paper. The paper contains kp value calculation however kinetic data showed a reduction in weight instead of mass gain, also the authors describe PBR as a solution for oxide scale formation however PBR shows some discrepancies with reality. I am rejecting the paper in this form.
Author Response

(The authors gave the same response as above.)

Reviewer 3 Report
Comments for authors:
The topic is interesting. The manuscript is well written in good English. The results are well described and adequately discussed. Therefore, just few minor corrections are recommended:
- Make the letters and numbers in Fig. 1 larger. They are too small in comparison with surrounded text. Especially in Fig. 1(c) some of them are barely readable. Why does not the fuel consumption correlate with the increasing thrust? The fuel curve is steeper. What is SFC? Please explain all in the text.
- While Fig. 1(a) is referred in the Introduction, other two (Figs. 1(b) and 1(c)) are not. All figures should be referred in related text.
- Line 93: Pay attention to typos. Mo solid solution is here written as “Moss”.
- Length bar scales are missing in Figs. 2 (a) and (b). It is always good to have a length scale embedded in the micrographs as it primary serve as etalon for estimation of the grain size.
- Line 106 and 108: “… Fig.3(g)” – There is no Fig.3(g)! Fig. 3 series ends at (f).
- Line 129: “3.3. XRD analysis…” There is no XRD analysis, this chapter contain only EDS results.
Author Response

(The authors gave the same response as above.)

Round 2
Reviewer 2 Report
Please see attachment

Author Response

(The authors gave the same response as above.)
